# Barrett’s Esophagus: An Updated Review

**DOI:** 10.3390/diagnostics13020321

**Published:** 2023-01-16

**Authors:** Peter M. Stawinski, Karolina N. Dziadkowiec, Lily A. Kuo, Juan Echavarria, Shreyas Saligram

**Affiliations:** Long School of Medicine, Health Science Center, University of Texas at San Antonio, 7703 Floyd Curl Dr., San Antonio, TX 78229, USA

**Keywords:** Barrett’s esophagus, esophageal adenocarcinoma, intestinal metaplasia, dysplasia, metaplasia

## Abstract

Barrett’s esophagus (BE) is a change in the distal esophageal mucosal lining, whereby metaplastic columnar epithelium replaces squamous epithelium of the esophagus. This change represents a pre-malignant mucosal transformation which has a known association with the development of esophageal adenocarcinoma. Gastroesophageal reflux disease is a risk factor for BE, other risk factors include patients who are Caucasian, age > 50 years, central obesity, tobacco use, history of peptic stricture and erosive gastritis. Screening for BE remains selective based on risk factors, a screening program in the general population is not routinely recommended. Diagnosis of BE is established with a combination of endoscopic recognition, targeted biopsies, and histologic confirmation of columnar metaplasia. We aim to provide a comprehensive review of the epidemiology, pathogenesis, screening and advanced techniques of detecting and eradicating Barrett’s esophagus.

## 1. Introduction

Barrett’s esophagus is a transformative process, in which metaplastic columnar epithelium replaces the endogenous stratified squamous epithelium in the lower portion of the esophagus [1]. This adaptation is a consequence of chronic gastroesophageal reflux disease (GERD) which damages the endogenous stratified squamous epithelium and over time has the potential to predispose to the development of adenocarcinoma of the esophagus [2]. It is estimated that 5.6% of adults in the United States have Barrett’s esophagus [3]. It is usually discovered during endoscopic examination of middle-aged and older adults; however, the majority of cases are unrecognized [2]. We believe there is a clinical need for an updated review article on BE, through this consolidated review of the literature, we hope to improve the awareness, diagnostic accuracy, detection, and eradication of esophageal lesions. We aim to review the most current understanding of pathophysiology, risks, and management of Barrett’s esophagus.

## 2. Pathophysiology

Barrett’s esophagus is believed to occur as a two-step process [4]. The first step, which occurs relatively quickly over a period of a few years, involves transformation of normal esophageal squamous mucosa into a simple columnar epithelium which lacks parietal cells, known as cardiac mucosa. This initial transformation is a result of exposure from chronic repeated episodes of refluxing gastric acid onto the squamous mucosa [4]. The presence of this initial transformation into a cardiac mucosa has been supported by objective markers of GERD, including incompetence of the lower esophageal sphincter (LES), increased acid exposure in the esophagus on 24-h pH monitoring, erosive esophagitis, and the presence of a hiatal hernia [5]. The precise molecular mechanism for this change is still unknown; however, Tobey et al. had shown that chronic exposure of esophageal mucosa to gastric acid, had produced increased intercellular spaces which allow for hydrochloric acid molecules to permeate down to the stem cells in the basal layer and stimulate afferent nerves [6]. The ability of gastric acid to permeate through, can be responsible for the sensation of heartburn and the phenotypic transformation to simple columnar epithelium.

The second step of intestinal metaplasia, which is thought to progress more slowly, over 5–10 years can progress due to combined genetic and environmental factors [7]. One pathway is known as gastric differentiation, which forms parietal cells within glands below the columnar mucosa through the expression of gastric genes [4]. This forms an oxynto-cardiac mucosa, which is not premalignant and favorable since it confers protection from further metaplasia. In the second pathway, the cardiac mucosa undergoes expression of intestinal genes causing the formation of goblet cells within the columnar mucosa, this is known as intestinal differentiation. Once detected on endoscopy and confirmed by histology, this meets the definition of Barrett’s esophagus [4]. Barrett’s esophagus has several factors which make it a premalignant mucosa, this includes an increased growth rate, decreased programmed cell death—apoptosis, and increased diploid and aneuploid cells compared to normal epithelium [8,9]. This set of factors accumulates and accelerates the process of cell dysplasia and are responsible for the malignant transformation which is observed in the progression of Barrett’s esophagus.

## 3. Molecular and Genetic Events Associated with the Development of Barrett’s Esophagus

The neoplastic progression in BE and esophageal adenocarcinoma (EAC) can be explained by several important molecular and genetic events, one of which is the loss of heterozygosity (LOH) of a gene. LOH is a chromosomal deletion of a whole gene in one chromosome, which requires reciprocal transcription to occur from the other adjacent, mutant, or inactivated gene [10]. The most common LOH in BE and EAC is in locus 9p21 and 17p13, involving genes CDKN2A and TP53. Inactivation of CDKN2A is believed to be the earliest inciting event of dysplasia and pathogenesis in BE [11]. TP53 is responsible for progression and accumulation of mutations, about 72–82.6% of TP53 mutations are identified in EAC, it is also suspected that mutations in TP53 are present before visible endoscopic detection of dysplasia [12].

## 4. Risk Factors Associated with the Development of Barrett’s Esophagus

Risk factors for Barrett’s esophagus in the general population vary widely based upon the studied population, from 0.4 to 20 percent overall worldwide. In the United States, Barrett’s esophagus shows the highest prevalence in White individuals over 50 years of age as compared to Hispanic or Asian descent, and lowest in Black individuals [13,14]. There are several factors that predispose to the development of Barrett’s esophagus, including GERD, peptic stricture, and erosive esophagitis—which confers a fivefold increased risk of Barrett’s esophagus at five-year follow up (relative risk ratio [RRR] 5.2, 95% CI 1.2–22.9) [15]. Central obesity is considered an independent risk factor for GERD, can pose a risk for Barrett’s esophagus, in patients with a body mass index (BMI) > 30 kg/m^2^ as compared to patients with (BMI) < 30 (odds ratio (OR) 1.4, 95% CI 1.1–1.6) [16]. Similar studies have found that a high waist-to-hip-ratio (≥0.9 in males and ≥0.85 in females) is associated with an increased risk of Barrett’s esophagus [17,18,19,20]. Not all patients who have these risk factors will develop Barrett’s Esophagus; further studies are needed to establish causal pathways of this condition. 

## 5. Screening for Barrett’s Esophagus

By definition, a screening test is the examination of a large population, with certain characteristics to detect a specific disease or disorder [21]. In this setting, screening for Barrett’s esophagus refers to an initial endoscopy that serves to identify individuals with Barrett’s esophagus, high-grade dysplasia, or esophageal adenocarcinoma (EAC). Screening programs can be effective for diseases which can be identified at early stages of progression, and which affect a significant proportion of the population. A screening program is only effective if it can be shown to significantly reduce the burden of disease and is cost effective. Although screening for Barrett’s esophagus in the general population is not routinely recommended, it may be considered in men with chronic (>5 years) and/or frequent (weekly or more) symptoms of gastroesophageal reflux and two or more risk factors [21]. These risk factors include age >50 years, Caucasian race, presence of metabolic syndrome, current or past history of smoking, and a confirmed family history in a first-degree relative of Barrett’s esophagus or EAC [22]. Although multiple societies suggest forms of selective screening in at-risk patient populations, this approach has never been validated in a prospective study [22,23,24,25]. Currently there are no data to suggest a decrease in mortality from EAC with entering patients into a Barrett’s esophagus screening program.

## 6. Diagnosis of Barrett’s Esophagus

The diagnosis of Barrett’s esophagus is a combination of several components, including recognition during endoscopy, appropriately targeted biopsies, and histologic confirmation of columnar metaplasia [26]. Endoscopic recognition of Barrett’s esophagus requires the presence of columnar epithelium ≥1 cm above the proximal margin of the gastric folds, based on the universally accepted Prague criteria. The Prague classification showed excellent concordance for length thresholds >1 cm but poor concordance for lengths <1 cm [27]. Current guidelines from American Gastroenterology Association (AGA) and American Society of Gastrointestinal Endoscopy (ASGE) do not explicitly provide a length threshold. As commonly seen in clinical practice, biopsies are obtained from the normal Z-line or a Z-line with <1 cm of variability. This has led to an overabundance and mislabeling of patients with Barrett’s esophagus who do not have the condition [27]. Ganz et al. had demonstrated 42 of 130 patients studied (32.3%) (95% CI, 24.4–41.1%) had been falsely diagnosed with Barrett’s esophagus. Based on the CI in this study, at least 24.4% of patients who carry a diagnosis of Barrett’s esophagus would be negative on repeat endoscopic examination [28].

Histologic confirmation of Barrett’s esophagus shows a combination of intestinalized columnar cells, gastric fundic and gastric cardia type cells present in the mucosa [29]. Harrison et al. demonstrated that the diagnostic yield of Barrett’s esophagus is maximized when at least eight biopsies are obtained [29]. The greater the overall length of columnar-lined epithelium, the higher the likelihood is the diagnostic yield of biopsies obtained from that segment.

## 7. Surveillance of Barrett’s Esophagus

Surveillance for Barrett’s esophagus takes careful consideration from both the patient and provider to help optimize outcomes and reduce mortality with esophageal adenocarcinoma. The goal of endoscopic surveillance is the early detection of dysplasia or carcinoma at an early and treatable stage. A patient-centered discussion which includes the timing of repeat endoscopy is important to help explain the risks and benefits before formally committing patients to a surveillance program. A 2009 systematic review which showed that patients with Barrett’s esophagus had a pronounced reduction in health-related quality of life and were at increased risk for major depressive disorder, anxiety disorder and stress [30]. Current Barrett’s esophagus surveillance guidelines are based largely on expert opinion from the three major gastrointestinal organizations in the United States, the American Gastroenterological Association (AGA), American College of Gastroenterology (ACG), and the American Society for Gastrointestinal Endoscopy (ASGE) [30].

Surveillance techniques should utilize high-resolution white light endoscopy to properly evaluate the mucosa in a thorough manner including insufflation of the lumen, retroflexion and inspection of the gastroesophageal junction. Gupta et al. described a correlation between EGD inspection times and positive detection of a segment of dysplasia [31].

Dysplasia is organized into a five-tier system that includes: negative for dysplasia, indefinite for dysplasia, low-grade dysplasia, high-grade dysplasia, and carcinoma. It is imperative that surveillance biopsies be performed only when active inflammation related to GERD is well controlled with antacid medications as this can result in false-positive results. The gold-standard for tissue sampling of Barrett’s esophagus is the Seattle Protocol, first described in 1993, which consists of four-quadrant biopsies at intervals of every 1–2 cm and separate samples of areas identified by mucosal irregularity along the entire involved segment [32,33].

If initial biopsies reveal no dysplasia, repeat endoscopy is recommended at intervals of 3 to 5 years depending on the length of the segment. The risk stratification based on segment length comes from a number of studies but notably a systematic review and meta-analysis of 20 studies found that increasing segment was associated with an increased risk of progression to HGD or EAC (OR 1.25; 95% CI 1.16–1.36; I^2^ = 45) [34,35]. Endoscopic surveillance for patients with <3 cm, short-segment Barrett’s esophagus is recommended at 5-year intervals. Patients with a segment length of ≥3 cm should undergo surveillance at 3-year intervals [35].

If the initial biopsies reveal indefinite for dysplasia, then antireflux therapy is optimized, often with a proton-pump-inhibitor to two times daily if the patient has not already been on high-dose therapy. A follow up endoscopy is carried out within 6 months [35]. If the repeat endoscopy results are indefinite dysplasia, then it is advised the patient to have a surveillance endoscopy every 12 months and if biopsy yields a diagnosis of no dysplasia or LGD/HGD then patients are treated using that algorithm [35].

If the initial biopsies reveal LGD, then it is recommended that this is confirmed by a second pathologist to ensure the LGD diagnosis is correct and to be certain that there is no HGD or EAC present. Endoscopic therapy is also suggested to reduce the risk of progression to HGD/EAC. Endoscopic surveillance is an acceptable option and if selected, is recommended at 6 months, 12 months, and then annually thereafter [35].

If biopsy initially reveals HGD or if known Barrett’s esophagus progresses to HGD then surveillance is no longer recommended and esophagectomy or endoscopic eradication therapy (EET) should be considered. A recent systematic review and meta-analysis demonstrated no difference between EET and esophagectomy regarding overall 1-, 3-, and 5-year survival and EAC mortality. However, lower rates of adverse events were noted in those undergoing EET compared with esophagectomy (RR 0.38, 95% CI 0.20–0.73) [36,37]. Available data suggest that most patients achieve complete eradication, within three endoscopy sessions [38].

## 8. Measurement of Barrett’s Esophagus

The Prague circumferential extent (the C value) and the maximum extent (the M value) classification criteria were developed to standardize the endoscopic grading system of Barrett’s esophagus. They originated as a set of criteria agreed upon by an international working group of researchers who used video-endoscopic recordings put together to create a standardized approach [39]. The Prague C and M classification requires the endoscopist to measure the nearest and farthest limits of the suspected abnormal mucosa. The proximal limit is set at the point of linear gastric mucosal folds, this indicates the gastroesophageal junction (GEJ) known as the origin point [40]. Columnar-lined epithelium is then measured to the circumferential extent (the C value) extending at least 1 cm above the GEJ, this area should not be biopsied as it may result in false positive results. The maximum extent (the M value) of the Prague Classification describes the length of the “tongues” measured from the GEJ [40].

## 9. Types of Barrett’s Esophagus

Barrett’s esophagus can be classified by many characteristics, including Barrett’s esophagus segment length, histologic type, and even genomic type. These classifications can provide stratification of risk for Barrett’s esophagus progression to esophageal adenocarcinoma. Simply, Barrett’s esophagus segment length can be defined as ultra-short (<5 mm), short (5 mm–3 cm), and long (>3 cm), measured by distance from the gastroesophageal junction to the Z-line [40].

The Prague C&M (circumferential and maximal) criteria are well-validated and widely accepted as a criteria system used to diagnose and classify Barrett’s esophagus. The distal extent of Barrett’s esophagus is defined as the proximal margin of the gastric mucosal folds. The circumferential segment (C) is measured from this distal margin to the proximal margin of the circumferential segment of Barrett’s esophagus. The maximal segment (M) is measured from the distal margin to the maximal extent of the proximally extending columnar mucosa. It should be noted that isolated segments or lesions with appearance of intestinal metaplasia are not measured or included in this classification system. The Prague system has been validated as reliable and reproducible [41].

Barrett’s histology can be differentiated into non-dysplastic and dysplastic types; dysplastic Barrett’s esophagus can be further graded as low-grade dysplasia (LGD) or high-grade dysplasia (HGD). Many classification systems have been proposed with varying validity. Narrow band imaging (NBI) has been used to classify Barrett’s lesions by endoscopic appearance by their mucosal and/or vascular patterns. In this system, dysplasia is identified by irregular mucosa and vascularity, while intestinal metaplasia is identified by an endoscopic appearance of ridging or villous pattern [42]. The Japan Esophageal Society (JES) classifies endoscopic vascularity by mucosal and vascular shape, categorizing them into either regular or irregular. The JES system showed promising diagnostic accuracy and reproducibility [43].

## 10. Importance of Confirmation of Diagnosis by a Second Pathologist

Because the histopathologic diagnosis of Barrett’s esophagus determines the type of management and level of surveillance, the accuracy of diagnosis is crucial for improving morbidity and mortality in Barrett’s esophagus and progression to EAC.

Several studies have shown significant intra-observer variability in the diagnosis of Barrett’s esophagus, particularly with indeterminate for dysplasia (IND) and low-grade dysplasia (LGD). In a study by Curvers, et al., it was demonstrated that 75% of patients diagnosed with LGD by community pathologists were downgraded to nondysplastic BE by expert GI pathologists at tertiary care centers [44].

There have also been studies demonstrating cases of LGD being mistaken for non-dysplastic or IND BE due to esophageal inflammation. Comparison of studies with high ratios of LGD to Barrett’s esophagus had lower incidence (2.15%) of EAC than studies with lower ratios which demonstrated higher incidence (5.7%) [45]. Given this variability, the incidence rate of progression to EAC for each type of dysplastic Barrett’s esophagus should be considered for pooled incidence rate alone.

## 11. Advanced Images to Detect Barrett’s Esophagus

Current standards recommend obtaining a high-quality endoscopic examination with careful inspection of the BE segment and adherence to the Seattle protocol for tissue sampling. Endoscopic surveillance of established BE can be potentially improved through adjunct advanced imaging during endoscopic evaluation. Advanced imaging technologies allow for in-depth visualization of abnormalities which may be overlooked during routine endoscopic evaluation. The majority of EAC found in BE is flat and nonpolypoid, causing it to be difficult to detect. Random biopsy is prone to sampling error as the distribution of affected areas is highly variable; small areas can be focal and may be easily missed. White light endoscopy inherently carries these limitations and data suggest that advanced imaging technologies have the potential to overcome and reduce these challenges.

### 11.1. Virtual Chromoendoscopy

Virtual chromoendoscopy uses technology built directly into the endoscope. The most widely used system is the Olympus narrow band imaging (NBI) (Figure 1) which applies a red–green–blue light filter to maximally highlight the surface mucosa and vascular pattern of the tissue being examined [46]. This narrower spectrum of 400–540 nm (compared to 400–700 nm white light) is matched to the absorption of hemoglobin, causing tissues such as blood vessels and blood to appear darker compared to the surrounding mucosa. Other widely used systems include the Fujinon Intelligent Color Enhancement (FICE) and the Pentax iScan which capture white light images and digitally process them to enhance the surface mucosa and vascular pattern [47]. A meta-analysis of nine studies showed a pooled sensitivity and specificity of greater than 94% each of detecting BE with NBI [48]. A separate study indicated an overall reduction in the number of biopsies while still detecting high-grade dysplasia (HGD) and early adenocarcinoma when white-light and NBI endoscopy were compared [49]. Virtual chromoendoscopy adds no cost, additional time or risk to the patient while providing a useful adjunct during routine endoscopy.

### 11.2. Volumetric Laser Chromoendoscopy (VLE)

Volumetric laser chromoendoscopy uses optical coherence tomography with infrared light to produce high-resolution, cross-sectional imaging of tissue in real-time without the need for contrast [50]. This technology can produce 1200 cross-sectional images during each 6 cm segment that is scanned, allowing larger BE segments to be evaluated in a shorter amount of time [51]. Each 6 cm segment can be scanned in approximately 90 s, providing mucosal and submucosal cross-sectional imaging with a resolution of 7 μm and a depth of 3 mm.

Experimental studies comparing VLE to endoscopically resected specimens have demonstrated sensitivities of 86–90% and specificities of 88–93% for the detection of dysplasia in BE [51]. In a large retrospective study comparing the detection of dysplasia in patients undergoing BE sampling with Seattle protocol biopsies, VLE without laser markings, and VLE with laser markings, both VLE with and without laser marking had statistically significant higher yield of dysplasia compared to Seattle protocol, (14% vs. 1%) and (11% vs. 1%), respectively [52]. Artificial intelligence (AI) technology is being developed and currently under study to aid in the interpretation of the abundance of information provided by VLE. This technology will help process collected images and highlight features that are associated with dysplasia, including hyper-reflective and hypo-reflective surfaces and a lack of layering in tissues. The benefits of VLE are that entire segments of BE can be imaged in a short period of time, abnormalities can be laser marked for targeting and it does not appear to increase endoscopic risk to patients.

## 12. Wide-Area Transepithelial Sampling (WATS)

Wide-area transepithelial sampling (WATS) is a three-dimensional (3D), computer-assisted technique which has been used as an adjunct to traditional forceps biopsy. The Seattle protocol has certain limitations, including sampling bias, limited sampling area, low compliance, and sampling variability among endoscopists. WATS uses endoscopic abrasive brush biopsy to sample transepithelial tissue circumferentially. The biopsy samples are then captured into histologic slices which are synthesized into a 3D image. The 3D imaging is analyzed by software algorithm. Images suspicious for dysplasia are then saved and evaluated by WATS-trained pathologist who interprets the data [53].

Adjunctive WATS has been shown to significantly increase diagnostic yield in BE screening and surveillance endoscopies over forceps biopsy. A large prospective multi-center study found that in Barrett’s screening, 1684 cases of 12,899 patients were diagnosed by forceps biopsy, whereas an additional 2570 cases were detected by WATS which were missed, yielding an increase in detection rate from 13.1 to 33% [53]. In the same study, dysplasia was diagnosed in 88 cases by forceps biopsy, and an additional 213 cases were diagnosed by WATS. Notably, these cases of dysplasia were classified into 35% low grade dysplasia, 60% indefinite for dysplasia, and 5% high grade dysplasia. 

Overall, forceps biopsy with WATS led to an absolute increase in detection of Barrett’s esophagus by 16% by pooled data from 11 studies [54]. Forceps biopsy alone diagnosed Barrett’s high-grade dysplasia or esophageal adenocarcinoma in 2.3% of patients, while forceps biopsy with WATS yielded an additional 2.1% of diagnoses [55]. In the same meta-analysis, the outcome of forceps biopsy alone was compared to WATS alone and found that WATS was negative in 62.5% of the cases of dysplastic Barrett’s diagnosed by forceps biopsy alone. This finding favors the idea of utilizing WATS as an adjunct rather than replacing forceps biopsy as a method of surveillance. 

## 13. Endoscopic Ultrasound (EUS)

In addition to white light endoscopy, endoscopic ultrasound (EUS) and endoscopic mucosal resection (EMR) (Figure 2) are often performed prior to endoscopic therapy of BE. EUS has been used to assess for submucosal invasion, given that the initial forceps biopsy does not width and depth of lesions. In early-stage neoplasia, EUS is also used to assess lymph node involvement. These results may help determine which patients are not candidates for endoscopic treatment and should instead be referred for surgical treatment. 

EUS was found to have poor sensitivity (50%), positive predictive value (40%), and 11% of patients were staged incorrectly, with 7% overstaged and 4% understaged, with EUS compared to EMR [56]. It was found that if staging with EUS alone, 7% of patients would have undergone unnecessary esophagectomy [56]. Pooled data yielded similarly low statistical detection of advanced disease by EUS; however, EUS identified advanced disease (HGD or EAC) in a significant 4% of non-nodular BE cases [57]. Overall, EUS had specificity of 89%, negative predictive value of 85%, and had an odds ratio of 12.6, meaning that it was 12.6 times more likely to show positive findings in patients with advanced disease than to show positive findings without advanced disease [57]. It has been discussed that the reason EUS may overclassify dysplasia because it can be difficult to ultrasonically differentiate between microscopic tumor invasion of tissue and peritumoral inflammatory changes. Nevertheless, it has been found that EUS remains an appropriate technique to assess for lymph node involvement prior to performing endoscopic treatment of advanced disease [58].

## 14. Role of Artificial Intelligence (AI) in Detection of Dysplasia in Barrett’s Esophagus

Over the last 20 years, there has been a tremendous increase in the development of artificial intelligence (AI) in the field of gastroenterology and hepatology, which heavily relies on imaging. Despite the fact that histopathologic specimen analysis remains the gold standard in establishing a diagnosis of BE, AI has enabled endoscopists to target specific lesions and rely less on random sampling [59].

Lesion detection by use of a computer algorithm made of 100 machine learned images from 44 BE patients, was one of the first studies to implement a detection algorithm [60]. The algorithm was able to diagnose neoplastic lesions with a sensitivity of 86% and specificity of 87% per patient [60]. Depth of invasion was tested in a study by Ebigbo et al., a deep learning system was trained to differentiate between T1a and T1b Barrett’s cancer using 230 white-light endoscopic images [61]. Results of this study showed a sensitivity of 77%, a specificity of 64% and accuracy of 71%, indicating how accurate prediction of submucosal invasion remains a challenge for both AI and experienced endoscopists [61].

Once AI becomes common use during upper endoscopy, the application will have the potential to detect suspicious areas of neoplasia, and measure lesion morphology and size. The endoscopist will receive alerts throughout the upper endoscopy procedure which can be decided if areas should be sampled or managed endoscopically with endoscopic resection [62,63]. The purpose behind AI is to further enhance and train physicians through human–computer collaboration rather than replacing human cognition [62,63]. As these systems become commercialized, consideration must be given to high upfront costs, but, in the long-term, it may prove to be cost-efficient if these systems maintain high-quality examination and detection performance. AI is a promising evolving area in gastroenterology with a vast potential to positively impact detection, diagnosis, and endoscopic treatment of BE.

## 15. Endoscopic Eradication Therapy (EET)

A non-dysplastic BE can progress to dysplastic BE and subsequently cancer. Given the progression of dysplastic BE to esophageal adenocarcinoma, advanced disease requires eradication therapy. Risk factors in BE patients include length of BE, presence of hiatal hernia, aneuploidy, and p53 loss of heterozygosity [64]. Cases of BE with visible mucosal abnormalities should be staged by endoscopic mucosal resection (EMR), which can also serve as eradication therapy. Classification of dysplasia, risk of distant involvement, and mucosal abnormalities determine need for locoregional versus surgical therapy.

## 16. Non-Dysplastic Barrett’s Esophagus (NDBE)

Cases of non-dysplastic Barrett’s esophagus have lower and slower progression to advanced disease. Non-dysplastic cases developed low-grade dysplasia (LGD) incidence of 4.3% per year [64]. It was found that 1 in 50 patients with non-dysplastic BE on multiple biopsies ended up progressing to high grade dysplasia at an interval of 6 years [64]. Pooled incidence of HGD and EAC from non-dysplastic BE was found to be 11.2% [65,66]^.^ Because of this low rate of incidence, non-dysplastic BE is does not require eradication therapy and is recommended for surveillance endoscopy alone every 3–5 years.

## 17. Low-Grade Dysplasia and Indefinite for Dysplasia

LGD is uniquely variable as a classification of BE, as many cases of LGD can be mistaken for non-dysplastic or “indefinite for dysplasia (IND)” BE in the presence of esophageal inflammation. In studies with higher ratio of LGD/BE, there was a reported lower incidence of EAC at 2.15%, while lower ratio was reported to have a higher incidence of EAC (5.7%) [64]. Combined incidence of HGD and EAC was found to be 1.73% annually [64]. This demonstrates how the high variability of LGD diagnosis should be confirmed with an expert GI pathologist before starting treatment. Patients with LGD should be on aggressive treatment with proton-pump inhibitors and evaluated with repeat endoscopy to confirm LGD. In a large multicenter study, 32% of patients who developed incidental LGD who were found to have intermittent LGD (between nondysplastic and LGD-BE) [46]. If repeat EGD demonstrates mucosal abnormalities or confirms LGD by histology, eradication therapy should be performed. Given moderate risk for progression of disease, careful surveillance is acceptable instead of eradication.

Insufficient data specific for recurrence of advanced disease following eradication therapy of LGD exist, so surveillance guidelines are currently recommended with overall dysplasia and EAC recurrence rate and disease-free intervals in mind [67]. Complete eradication of LGD should have surveillance endoscopy 6 months following treatment and annually thereafter.

## 18. High-Grade Dysplasia

HGD has the highest risk of progression to EAC, found to have a weighted EAC incidence rate of 6.58 per 100 patient years in patients undergoing regular surveillance, not normalizing for BE segment length [68]. Initial grade of dysplasia had the strongest prognostic factor and development of EAC. Endoscopic eradication therapy, including EMR and/or ablation therapy, is indicated in HGD given high risk of progression and has been shown to have achieved complete eradication in 83.7% of patients requiring a median of two treatment sessions [68]. Surveillance following eradication is informed by recurrence rate, which has been found to be 13.5% following treatment in the same study. Because of recurrence rate and risk of progressive disease, surveillance is currently recommended more frequently (3–6 months) in the first few years following therapy, and annually thereafter. Factors such as BE segment length have been associated with longer treatment to reach eradication and higher recurrence rate following eradication therapy and may be considered in an individualized surveillance protocol [69,70].

## 19. Intramucosal (T1a) Cancer

Esophageal carcinoma is staged using the TNM (tumor-node-metastasis) staging system. Early stage EAC is divided by depth of invasion, T1a (mucosal) and T1b (submucosal). Because EAC prognosis is generally poor, treatment is indicated even in early T1a stage cancer.

Given that lymph node metastasis in early T1a stage esophageal carcinoma is low (ranging from 0–1.8%), the effective and safe treatment of choice is endoscopic eradication therapy [70]. Surgical resection, or esophagectomy, is associated with higher risk of morbidity and mortality. The efficacy of various endoscopic methods is comparable, as detailed below, and should be selected depending on lesion width and depth of invasion [71]. Histology of T1a cancer can be further separated into moderately differentiated versus poorly differentiated cancer. Poorly differentiated cancers are defined as high-risk tumors which have lymphovascular metastasis rate, ranging greatly from 8.7–42% [71]. Given this risk, poorly differentiated T1a cancer should be treated with surgical resection, as the risk of esophagectomy outweighs mortality risk associated with metastasis.

## 20. Submucosal (T1b) Cancer

Submucosal (T1b) cancer has been associated with higher rates of lymph node metastases; however, this rate reportedly varies by several studies depending on the depth of submucosal invasion. This submucosal invasion can be divided into three layers (sm1–sm3), sm3 being the upper third layer of the submucosa. It has been found that endoscopic treatment is effective in complete eradication in 87% of patients with T1b-sm1, and further improved to 97% in focal disease <2 cm [72]. However, this was not confirmed in another large study, which found that endoscopic resection was suboptimal in T1b cancer with muscularis mucosal invasion compared to surgical resection in adequately resection and assessment of lymph node metastases [73].

Given confounding results, patients with T1b EAC can be approached by careful assessment of submucosal invasion and assessment of lymphovascular involvement in imaging by PET, CT, and/or EUS.

## 21. T2 Cancer

Locally advanced esophageal cancer which invades the muscularis propria (T2) is aggressive, characterized by poor 5-year survival, ranging from 17.1% to 23% even in patients who received treatment [74]. Currently, T2 should be managed with neoadjuvant chemoradiotherapy plus surgical resection which shows survival benefit over treatment by surgery alone; survival rates at 5 years for each has been found to be 47% and 33%, respectively [75].

## 22. Treatment Modalities

Several treatment modalities for advanced disease and cancer have been developed and studied to effectively treat disease, from endoscopic resection and ablation to more aggressive surgical options. Some of the well-studied endoscopic treatment modalities are detailed below.

### 22.1. Endoscopic Mucosal Resection (EMR)

Endoscopic mucosal resection (EMR) is a widely used technique for diagnosing and eradicating superficial BE dysplasia and neoplasia. This technique involves endoscopic evaluation of mucosal tissue, followed by targeted resection of visible mucosal abnormalities suggestive of dysplasia. The resected tissue is sent for histopathological study to confirm diagnosis and staging. For well-differentiated dysplasia without lymphovascular invasion, EMR can be considered curative if the lesion is superficial and resection margins are negative [76,77]. For cases of circumferential, short segment BE, EMR may be a staged procedure and be completed in more than one session.

The pooled efficacy of patients who had complete eradication of intestinal metaplasia (CE-IM) was 79.6% and complete eradication of neoplasia (CE-N) was 94.9%, with substantial heterogeneity (I^2^ > 25%) [78]. There is risk of recurrence which is associated with fragmented resection technique and longer BE segment [76]. Pooled recurrence rate for EMR was 0.7% for EAC, 3.3% for dysplasia, and 12.1% for intestinal metaplasia [78].

### 22.2. Endoscopic Submucosal Dissection (ESD)

Endoscopic submucosal dissection (ESD) is a technique, initially created for resection of gastric tumors, which utilizes en bloc resection. This modality aims to remove entire lesions, accommodating for varying widths and depths. Though more time consuming and technically demanding than EMR, it was shown in a small prospective randomized control trial that direct comparison that the rate of complete resection was significantly higher (58.8%) following ESD than EMR (11.8%) following a single session (*p* = 0.03) [79]. Pooled data demonstrated ESD cure rate of 92.3% compared to that of EMR with 52.7% cure rate (*p* < 0.001) and found significantly lower recurrence rate in ESD (0.3%) compared to EMR (11.5%) (*p* < 0.001) [80]. For the same reason, ESD are being used to treat submucosal (T1b-Sm1) lesions. More large population data are needed to assess for CE-IM and CE-N of ESD. Procedure time may be a limiting factor, as the mean EMR time was 36.7 min while the mean ESD time was 83.3 minutes [80].

Though efficacy has been shown in several studies, ESD has not yet been widely accepted as preferred eradication therapy likely because the technique is more technically demanding and time consuming than other endoscopic eradication modalities.

### 22.3. Radiofrequency Ablation (RFA)

Radiofrequency ablation (RFA) (Figure 3) can be employed to eradicate circumferential areas of dysplastic BE. It is often used in conjunction with EMR for complete diagnosis and treatment; nodular BE requiring EMR for appropriate resection and non-nodular BE benefiting from targeted or focal ablation. When required for complete eradication following initial EMR or EMR with RFA, serial RFA is typically performed every 3 months [81,82].

RFA has been shown to be highly effective in completely eradicating intestinal metaplasia and all grades of dysplasia and neoplasia, and on average required 3–4 treatment sessions for eradication [81]. Pooled CE-IM rates of 78% and rate of complete eradication of dysplasia (CE-D) of 91% [82]. It was found that pre-treatment histology was a non-significant determining factor in efficacy, with LGD more likely to demonstrate CE-IM and CE-D than HGD and/or intramucosal cancer [82]. Characteristics associated with incomplete eradication included BE segment length, incomplete healing between serial RFA therapy, and requirement of more treatment sessions [81]. RFA treatment failure can be difficult to define by threshold number of treatment sessions given that cases have been successfully eradicated following more than four or five sessions. RFA is widely accepted as first-line therapy given efficacy and safety; however, adverse effects can include, most commonly, strictures, bleeding, and pain.

### 22.4. Cryotherapy

Cryotherapy involves cold-temperature ablation and can be performed via various techniques, from liquid nitrogen spray to compressed carbon dioxide to cryo-balloon therapy [83]. Employment of cryotherapy is monitored endoscopically, and dosing is measured by freeze time and cycle count. Endoscopic follow-up and as needed serial cryotherapy is performed approximately every 3 months.

Pooled efficacy of various cryotherapy modalities demonstrated CE-IM of 64.2% and CE-D of 84.8% with substantial heterogeneity, with median number of four treatment sessions. Several studies identified persistent intestinal metaplasia and dysplasia, with pooled rate of 13.7% and 7.3, respectively [84]. Theoretically, cryotherapy offers advantages over RFA by resulting in less destruction of tissue architecture and utilizing cold therapy, resulting in a lesser side effect profile in terms of stricture development and post-procedural pain; however, this has not been well-documented in head-to-head comparison.

### 22.5. Hybrid Argon Plasma Coagulation (APC)

Hybrid argon plasma coagulation (APC) ablation is a technique which uses the combination of a submucosal injection of saline followed by APC ablation. This hybrid technique was developed to improve tolerability and decrease risk of stricture formation, the most commonly occurring adverse effect of RFA and other thermal ablation procedures [85]. Repeat hybrid APC treatment sessions have been performed at 6–12 weeks follow-up with biopsies to assess for successful eradication [34].

In a prospective, multi-center study, it was found that CE-IM was achieved in 88.4% of hybrid APC cases and CE-N was achieved in 98% [86]. Treatment failure in the study was defined by no visible regression of BE or continued positive biopsies for dysplasia or neoplasia. Adverse effects were found to be moderate for strictures in the study; however, there are no pooled data to compare hybrid APC outcomes to other forms of endoscopic ablation.

### 22.6. Endoscopic Surveillance after Achieving Complete Eradication of Intestinal Metaplasia

Recurrence of intestinal metaplasia after successful endoscopic eradication therapy occurs in 8–10% of patients yearly, most commonly at the gastroesophageal junction (74%) and in the tubular esophagus (26%) [87]. In addition, the timing of recurrence detection is reported in the first year after complete eradication of intestinal metaplasia [88]. Sawas et al. concluded that recurrence in the first year is likely due to incompletely treated disease rather than recurrent disease, justifying the need for increased surveillance in the first year [88]. In a large multicenter, cohort study by Sami et al. the recurrence rates in non-dysplastic BE, low-grade or high-grade dysplasia or carcinoma after RFA did not plateau or decrease at initial endoscopic follow up- supporting continued yearly surveillance [89]. Current guidelines base surveillance intervals on pre-ablation histology. In patients with low-grade dysplasia preablation, surveillance is recommended at 1 year after complete eradication of intestinal metaplasia and every 2 years thereafter [89]. Patients with high-grade dysplasia and intramucosal carcinoma after complete eradication are to be screened at 3 months, 6 months, and 12 months and then annually thereafter [90].

Rarely, endoscopic ablation of BE can bury metaplastic glands that contain neoplastic potential, this is defined as the presence of lamina propria in the squamous biopsy. The frequency of which ranges from 12.9% to 91% after RFA [91]. The associated cancer risk with buried metaplasia is largely unclear. For patients treated with RFA, the frequency of buried metaplasia is low; however, this may be explained by the long length of time required for lesions to evolve into neoplasia [90]. The clinical relevance of buried metaplasia remains questionable due to limited research into its malignant potential, with further studies required to determine appropriate screening intervals [91].

## 23. Treatment of Refractory Barrett’s Esophagus

Although endoscopic eradication therapies as described above are very successful, there is a population of patients in whom complete eradication of Barrett’s esophagus cannot be achieved. In a multi-center prospective trial in 2013 identified several risk factors associated with incomplete eradication including scar regeneration with Barrett’s epithelium resection, active reflux esophagitis, narrow esophageal diameters, and longer years of dysplasia presence [92].

Several modifiable factors should be fully optimized in order to achieve complete eradication in addition to endoscopic eradication therapy. In those patients with incomplete eradication, further titration of their PPI regimen is imperative. Sometimes, despite optimal PPI therapy, acid reflux persists and incomplete eradication results that is refractory to EET; these patients may be considered for antireflux surgery such as Nissen fundoplication. Skrobić et al. evaluated patients who had refractory Barrett’s esophagus after either staying on PPI therapy or those who underwent Nissen fundoplication and found recurrence in 20% of patients on PPI, whereas recurrence occurred only in 9% of patients who had undergone fundoplication [93]. For patients who are not candidates for surgical anti-reflux therapy, endoscopic anti-reflux procedures may be an option such as transoral incisionless fundoplication (TIF). In refractory Barrett’s esophagus is the EndoRotor^®^ Endoscopic Resection System which is an automated mechanical non-thermal endoscopic resection system that can be applied in refractory cases to dissect tissue from peripheral margins. Currently there is a prospective randomized study which evaluates EndoRotor versus continued ablation for refractory Barrett’s [91,94].

## 24. Conclusions

Only a small group of patients identified to have Barrett’s esophagus go on to develop EAC. However, the particularly high mortality and cost to the healthcare system propels the screening, surveillance, and treatment practices of Barrett’s esophagus, with the ultimate goal of preventing advanced neoplasia. The incidence rates of EAC are rising which reinforces the need for improved screening and surveillance practices. The development of cost-effective screening tools and technical advances in detection of metaplasia and neoplasia is imperative to identify those at risk. The area of Barrett’s esophagus therapeutic management is rapidly evolving. Endoscopic eradication therapies have been shown to be effective and new therapies continue to be developed as advances are made.

## Figures and Tables

**Figure 1 diagnostics-13-00321-f001:**
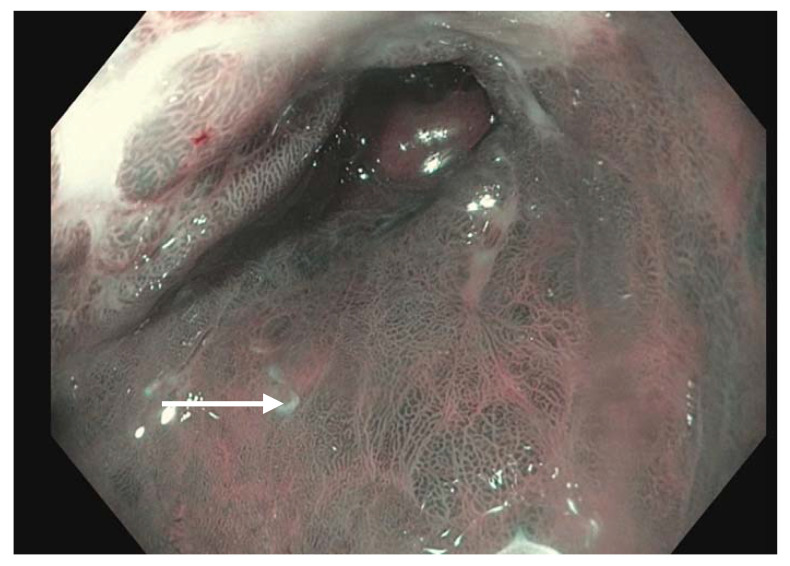
Narrow band imaging applied to a segment of Barrett’s esophagus (white arrow).

**Figure 2 diagnostics-13-00321-f002:**
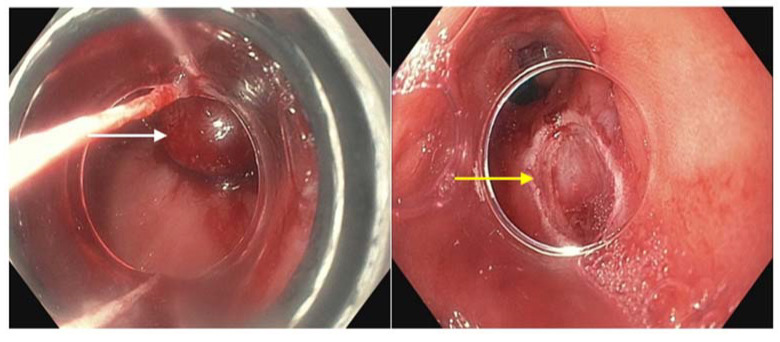
Endoscopic mucosal resection of a dysplastic segment ((**left**), white arrow) and exposed area of submucosa following EMR treatment, also known as a post-EMR defect ((**right**), yellow arrow).

**Figure 3 diagnostics-13-00321-f003:**
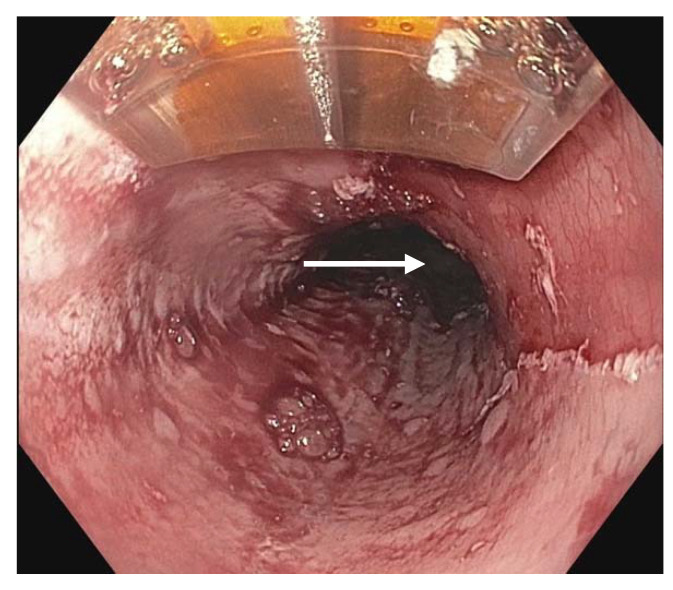
Radiofrequency ablation being applied using a probe toa dysplastic mucosal area (white arrow).

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
