# Peer review of "Barrett’s Esophagus: An Updated Review"

_diagnostics, 2023, doi:10.3390/diagnostics13020321_

Round 1

Reviewer 1 Report

Congratulations to the authors on this brilliant review. Stawinski et al. performed a comprehensive review of the pathophysiology, clinical and imaging evaluation, risk stratification, and management of Barrett's esophagus. I have some remarks.

Abstract: The study lacks an Abstract

Keywords: The study lacks keywords.

Body of the manuscript: There are several statements lacking references. For example: "Barrett’s Esophagus is a condition in which metaplastic columnar epithelium re- 21 places the endogenous stratified squamous epithelium in the lower portion of the esophagus."; "Although screening for Barrett’s esophagus in the general population is not routinely recommended, it may be considered in men with chronic (>5 years) and/or frequent 84 (weekly or more) symptoms of gastroesophageal reflux and two or more risk factors."; "Histologic confirmation of Barrett’s Esophagus shows a combination of intestinalized 108 columnar cells, gastric fundic and gastric cardia type cells present in the mucosa."

Introduction: In the Introduction section, the authors should provide a justification of the article's importance for the readers. Why do they think there is a need for an updated review of Barrett's esophagus?

Although review manuscripts usually do not present any Methodology section, it is reasonable to provide some insights into how the study's selection was performed. The SANRA (a scale for the quality assessment of narrative review articles) states that a convincing narrative review will be transparent about the sources of information on which the text was based. The search methods should not be as rigorous as a systematic review, but it is necessary to specify search terms and types of literature searched. I would also recommend informing the databases searched. 

Reference: Baethge C, Goldbeck-Wood S, Mertens S. SANRA—a scale for the quality assessment of narrative review articles. Research integrity and peer review. 2019 Dec;4(1):1-7. 

Author Response

Abstract: The study lacks an Abstract

Barrett’s esophagus (BE) is a change in the distal esophageal mucosal lining, whereby metaplastic co-lumnar epithelium replaces squamous epithelium of the esophagus. This change represents a pre-malignant mucosal transformation which has a known association with the development of esophageal adenocarcinoma. Gastroesophageal reflux disease is a risk factor for BE, other risk factors include patients who are Caucasian, age > 50 years, central obesity, tobacco use, history of peptic stricture and erosive gastritis. Screening for BE remains selective based on risk factors, a screening program in the general population is not routinely recommended. Diagnosis of BE is established with a combination of endoscopic recognition, targeted biopsies, and histologic confirmation of columnar metaplasia. We aim to provide a comprehensive review of the epidemiology, pathogenesis, screening and advanced techniques of detecting and eradicating Barrett’s esophagus.

Keywords: The study lacks keywords.

 Barrett's esophagus; Esophageal adenocarcinoma; Intestinal metaplasia; Dysplasia; Metaplasia

Body of the manuscript: There are several statements lacking references. For example: "Barrett’s Esophagus is a condition in which metaplastic columnar epithelium re- 21 places the endogenous stratified squamous epithelium in the lower portion of the esophagus."; "Although screening for Barrett’s esophagus in the general population is not routinely recommended, it may be considered in men with chronic (>5 years) and/or frequent 84 (weekly or more) symptoms of gastroesophageal reflux and two or more risk factors."; "Histologic confirmation of Barrett’s Esophagus shows a combination of intestinalized 108 columnar cells, gastric fundic and gastric cardia type cells present in the mucosa."

These references were added into the paper.

Introduction: In the Introduction section, the authors should provide a justification of the article's importance for the readers. Why do they think there is a need for an updated review of Barrett's esophagus?

 We believe there is a clinical need for an updated review article on BE, through this consolidated review of the literature, we hope to improve the awareness, diagnostic accuracy, detection, and eradication of esophageal lesions.

Although review manuscripts usually do not present any Methodology section, it is reasonable to provide some insights into how the study's selection was performed. The SANRA (a scale for the quality assessment of narrative review articles) states that a convincing narrative review will be transparent about the sources of information on which the text was based. The search methods should not be as rigorous as a systematic review, but it is necessary to specify search terms and types of literature searched. I would also recommend informing the databases searched. 

 The Pubmed database was used as the primary source for literature review. Keywords as stated above were used. The types of literature included in this article include the most recent narrative manuscripts, systematic reviews and review of guidelines on Barrett’s esophagus diagnosis and management. Key statements were backed by references and cited appropriately. SANRA-a scale for the quality assessment of narrative review articles was applied to self-appraise the article by the authors of this manuscript to ensure a high-quality review of Barrett’s esophagus.

Reference: Baethge C, Goldbeck-Wood S, Mertens S. SANRA—a scale for the quality assessment of narrative review articles. Research integrity and peer review. 2019 Dec;4(1):1-7. 

This reference was added into the paper

Reviewer 2 Report

In this manuscript, the authors have provided a collection of information on the pathophysiology, risks, and management of Barrett’s esophagus. The manuscript is well-written. However, the following points should be addressed to improve the manuscript.

1. The authors should provide an abstract.

2. A section on the molecular markers of Barrett's esophagus is needed.

Author Response

  1. The authors should provide an abstract.

Barrett’s esophagus (BE) is a change in the distal esophageal mucosal lining, whereby metaplastic co-lumnar epithelium replaces squamous epithelium of the esophagus. This change represents a pre-malignant mucosal transformation which has a known association with the development of esophageal adenocarcinoma. Gastroesophageal reflux disease is a risk factor for BE, other risk factors include patients who are Caucasian, age > 50 years, central obesity, tobacco use, history of peptic stricture and erosive gastritis. Screening for BE remains selective based on risk factors, a screening program in the general population is not routinely recommended. Diagnosis of BE is established with a combination of endoscopic recognition, targeted biopsies, and histologic confirmation of columnar metaplasia. We aim to provide a comprehensive review of the epidemiology, pathogenesis, screening and advanced techniques of detecting and eradicating Barrett’s esophagus.

  1. A section on the molecular markers of Barrett's esophagus is needed.

The neoplastic progression in BE and esophageal adenocarcinoma (EAC) can be explained by several important molecular and genetic events, one of which is the loss of heterozygosity (LOH) of a gene. LOH is a chromosomal deletion of a whole gene in one chromosome, which requires reciprocal transcription to occur from the other adjacent, mutant, or inactivated gene1. The most common LOH in BE and EAC is in locus 9p21 and 17p13, involving genes CDKN2A and TP53. Inactivation of CDKN2A is believed to be the earliest inciting event of dysplasia and pathogenesis in BE2. TP53 is responsible for progression and accumulation of mutations, about 72-82.6% of TP53 mutations are identified in EAC, it is also suspected that mutations in TP53 are present before visible endoscopic detection of dysplasia3.

  1. Maslyonkina KS, Konyukova AK, Alexeeva DY, Sinelnikov MY, Mikhaleva LM. Barrett's esophagus: The pathomorphological and molecular genetic keystones of neoplastic progression. Cancer Med. 2022 Jan;11(2):447-478. doi: 10.1002/cam4.4447. Epub 2021 Dec 6. PMID: 34870375; PMCID: PMC8729054.
  2. Reid BJ, Barrett MT, Galipeau PC, et al. Barrett's esophagus: ordering the events that lead to cancer. Eur J Cancer Prev. 1996;5(Suppl 2):57‐65.
  3. Dulak AM, Stojanov P, Peng S, et al. Exome and whole‐genome sequencing of esophageal adenocarcinoma identifies recurrent driver events and mutational complexity. Nat Genet. 2013;45(5):478‐486.